# Mimicking reductive dehalogenases for efficient electrocatalytic water dechlorination

Yuan Min[1], Shu-Chuan Mei[1], Xiao-Qiang Pan[1], Jie-Jie Chen [1]✉, Han-Qing Yu [1]✉ & Yujie Xiong [2]✉

Electrochemical technology is a robust approach to removing toxic and persistent chlorinated organic pollutants from water; however, it remains a challenge to design electrocatalysts with high activity and selectivity as elaborately as natural reductive dehalogenases. Here we report the design of high-performance electrocatalysts toward water dechlorination by mimicking the binding pocket configuration and catalytic center of reductive dehalogenases. Specifically, our designed electrocatalyst is an assembled heterostructure by sandwiching a molecular catalyst into the interlayers of two-dimensional graphene oxide. The electrocatalyst exhibits excellent dechlorination performance, which enhances reduction of intermediate dichloroacetic acid by 7.8 folds against that without sandwich configuration and can selectively generate monochloro-groups from trichloro-groups. Molecular simulations suggest that the sandwiched inner space plays an essential role in tuning solvation shell, altering protonation state and facilitating carbon−chlorine bond cleavage. This work demonstrates the concept of mimicking natural reductive dehalogenases toward the sustainable treatment of organohalogen-contaminated water and wastewater.

Extensive research efforts have been made to eliminate the threat of chlorinated organic pollutants to human health by developing constantly updated technologies for water dechlorination[1–4]. Water dechlorination usually involves complicated multi-step processes, including pollutant adsorption, detoxification, and subsequent catalytic degradation or separation[5,6]. The existing chemical dechlorination scenarios, which are usually energy intensive, are limited by the competing reactions and often cause secondary pollution[7]. To over this limitation, electrochemical technology has been reported as an effective approach for the elimination of various organic pollutants[8,9]. In particular, electroreduction offers an attractive method to improve the activity and selectivity of catalysts for triggering target reactions via multiple proton-coupled electrons transfer[10–12]. The electro-

reductive dehalogenation of water containing organochlorinated pollutants ensures the removal of chlorine under mild conditions[13,14]. Nevertheless, it still remains a challenge to design electrocatalysts with high activity and selectivity as elaborately as natural reductive dehalogenases.

The ubiquitous reductive dehalogenases in the environment play critical roles in dehalogenation by using organohalides as terminal electron acceptors in microbial respiratory metabolism[15,16]. To date, all the purified reductive dehalogenases contain cobalamin cofactor (i.e., vitamin $B_{12}$ and its derivatives). For instance, the Co metal center of a dehalogenase homologue (NprdhA) of *Nitratireductor pacificus* pht-3B can be directly bonded to the halogen on the aromatic ring to cleave the carbon−halogen bond[17]. The tetrachloroethylene reductive

[1]Department of Environmental Science and Engineering, University of Science and Technology of China, Hefei, Anhui 230026, China. [2]Hefei National Research Center for Physical Sciences at the Microscale, Collaborative Innovative Center of Chemistry for Energy Materials (iChEM), School of Chemistry and Materials Science, University of Science and Technology of China, Hefei, Anhui 230026, China. ✉e-mail: chenjiej@ustc.edu.cn; hqyu@ustc.edu.cn; yjxiong@ustc.edu.cn

dehalogenase (PceA) of *Sulfurospirillum multivorans* achieves dehalogenation via long-range electron transfer mechanism on Co center coupled with neighboring residues[18]. Looking into the working mechanisms of reductive dehalogenases, we can gain some clues for the design of electrocatalysts toward electrochemical water dechlorination. In the systems of reductive dehalogenases, biological reductions occur at the liquid−enzyme interface, i.e., at the catalytic center associated with the substrate-binding pocket formed by the residues for the supply of hydrogen bonds, involving controllable transfer of protons and electrons[19–22]. In light of this aspect, the development of an enzyme-like liquid−electrode interface is highly desired for the selective electroreduction of specific pollutants.

We thus envision that mimicking microbial reductive dehalogenases should be a feasible strategy for enhancing the performance of electrochemical reduction technology in the removal of organochlorinated pollutants. To this end, two-dimensional (2D) materials with tunable compositions and interlayers may provide a platform that offers the opportunities of rationally arranging active sites and controlling catalytic microenvironment to ensure reaction rate and selectivity[23]. Great efforts have been devoted to improving the electrochemical performance with a focus on the surface of 2D materials by creating heterostructures with different materials, exposing edge sites, and doping heteroatoms[24]. Despite the construction of 2D heterostructures, the unique functions of inner space, like the protein binding pocket, in electron transfer or proton transport remain largely unexplored. To mimic the reductive dehalogenases, the inner space of 2D heterostructures, in which guest species can be intercalated, should be fully utilized. Graphene and its oxide (GO), well-known 2D materials for long-range π conjugation[25] and special anisotropy[26], have served as a useful platform for intercalating guest species into interlayer space[27,28]. Most of the catalytic centers in enzymes have similar functions to molecular catalysts, which can be readily intercalated into the interlayer space to suppress the commonly observed swelling of 2D layers in liquid solvents by noncovalent interactions and enhance the overall stability[29]. As such, 2D materials intercalated with molecular catalysts with precise active sites for selective catalysis and suitable inner space for binding pocket-like configuration have an immense potential for electrochemical applications.

Here we designed and fabricated a universal framework of a protein-binding pocket-like heterostructure with GO toward the application of electrocatalytic water dechlorination. The controllable inner space coupled with tailored molecular catalysts enabled boosting the electroreduction of substrates. To achieve reductive dechlorination of sodium trichloroacetate (TCA), the catalytic center (vitamin $B_{12}$) of dehalogenase in organohalide-respiring bacteria was chosen as the molecular catalyst to construct the GO-$B_{12}$-GO heterostructure. The electrochemical dechlorination capability of this sandwich-like configuration surpassed the electrocatalysts in other forms. The mechanisms of the inner space tuning the electron and proton transfer to enhance the electrochemical performance were elucidated by employing a suite of electrochemical characterizations, density functional theory (DFT) calculations and molecular dynamics (MD) simulations. To demonstrate the universality of our concept, nitrogen-doped graphene and cobalt phthalocyanine (CoPc) were further adopted as alternative 2D material and cobalt-centered catalyst for dechlorination. This work provides a strategy for designing high-performance electrodes toward electrocatalytic water purification, by mimicking the binding pocket configuration and active sites of natural enzymes.

## Results

### Binding pocket-like configurations of electrodes

To design and fabricate the binding pocket-like electrocatalyst, the structure, and function of the $B_{12}$-dependent dehalogenase were fully dissected. Microbial reductive dechlorination occurs at the binding

pocket of dehalogenase, including the cofactor ($B_{12}$) and the surrounding key residues with aromatic rings and hydroxyl groups[18]. Substrates with carbon−chlorine (C−Cl) bonds are the final electron acceptors of microbial organohalide respiration. The subsequent step is the formation of C−H bonds through proton transfer. Unlike bulk water, the hydrophobic inner space of dehalogenase provides a local environment for proton transfer through a narrow channel. Limited by the channel, the confined water molecules are able to stabilize the frustrated hydrogen-bond networks with the groups of $B_{12}$−$CONH_2$ and −OH of the surrounding aromatic residues (Fig. 1a). To probe the roles of $B_{12}$ as a molecular catalyst and the steric effect of inner space on reductive dechlorination, an electrochemical system with three configurations of cathodes was adopted: the GO electrode without $B_{12}$, the GO-$B_{12}$ electrode with surface-adsorbed $B_{12}$, and the GO-$B_{12}$-GO electrode with additional intercalated $B_{12}$.

MD simulations were employed to visualize the electrode assembly processes (Fig. 1b, Supplementary Fig. 1–4 and Supplementary Note 1). The changing profiles of interlayer distance (Supplementary Fig. 5) suggest the stacking behavior of GO sheets, simultaneously driven by the Van der Waals (vdW) forces of GO sheets, the hydrogen bonding of $B_{12}$-$CONH_2$ and the solvation effect of water. At the equilibrium stage, the average interlayer distance of GO-$B_{12}$ was similar to that of the GO electrode but smaller than that of the sandwich-like GO-$B_{12}$-GO electrode. Moreover, the layered structures were confirmed by synchrotron-radiation photoelectron spectroscopy (SRPES) measurements with the depth profiling of GO, GO-$B_{12}$, and GO-$B_{12}$-GO electrodes (Supplementary Fig. 6–8 and Supplementary Note 2). X-ray absorption fine structure (XAFS) spectroscopy was used to identify the R space for GO-$B_{12}$-GO, giving a dominant Co−N coordination at 1.88 Å (Fig. 1c, d). Moreover, no characteristic peaks for Co−Co contribution at higher R values could be found, indicating that the isolated cobalt atoms were distributed on GO-$B_{12}$-GO. The quantitative coordination configuration of the Co atom in the catalyst was acquired by EXAFS fitting (Supplementary Fig. 9 and Supplementary Table 1), suggesting a coordination number (CN) of 3.7 for Co−N. Aberration-corrected high-angle annular dark-field scanning transmission electron microscopy (HAADF-STEM) images further illustrate the uniformly dispersed Co species with no observable Co nanoparticles in the GO-$B_{12}$-GO (Fig. 1e). Additionally, electron energy-loss spectroscopy (EELS) analysis shows the homogeneous distribution of C, N and Co elements over the entire domain. High-resolution X-ray photoelectron spectroscopy (XPS) was further employed to examine the surface groups of GO. For the C 1$s$ spectrum, the characteristic peaks at 284.8, 286.9 and 288.6 eV are assignable to C−C, C−O, and C=O, respectively (Supplementary Fig. 10a). The O 1$s$ peaks with binding energies of 531.5, 532.3, 533.0, and 533.8 eV represent O−C=O, C=O, C−OH, and C−O−C, respectively (Supplementary Fig. 10b).

### Structural characterizations of electrodes

The composition and configuration of the electrodes were controlled using a cyclic drop−dry process followed by electrochemical activation (Fig. 2a). Electrochemical reaction kinetics measurements and grazing incidence X-ray diffraction (GIXRD) studies were performed to provide insights into the structural characteristics of the electrodes. Multi-scan cyclic voltammetry (CV) experiments were conducted in a potential range of 0.62 to 0.86 V (*vs*. normal hydrogen electrode, NHE), as shown in Supplementary Fig. 11, giving a normalized capacitance of GO-$B_{12}$-GO of 40 μF cm$^{-2}$ (Fig. 2b), which was differentiated by the equation $i(V) = k_m v + k_n v^{1/2}$ to analyze the surface-determined process ($k_m$) or the diffusion-controlled process ($k_n$)[30]. Compared with GO-$B_{12}$-GO, the higher diffusion contribution of GO alone (93%) was ascribed to the smaller distance between Na$^+$ ion and inner surface of electrode[31,32]. As a result, the interlayer distance of the GO-$B_{12}$-GO electrode was larger than that of GO, which could be examined by GIXRD profiles. The GO electrode had a diffraction peak at 10.047°

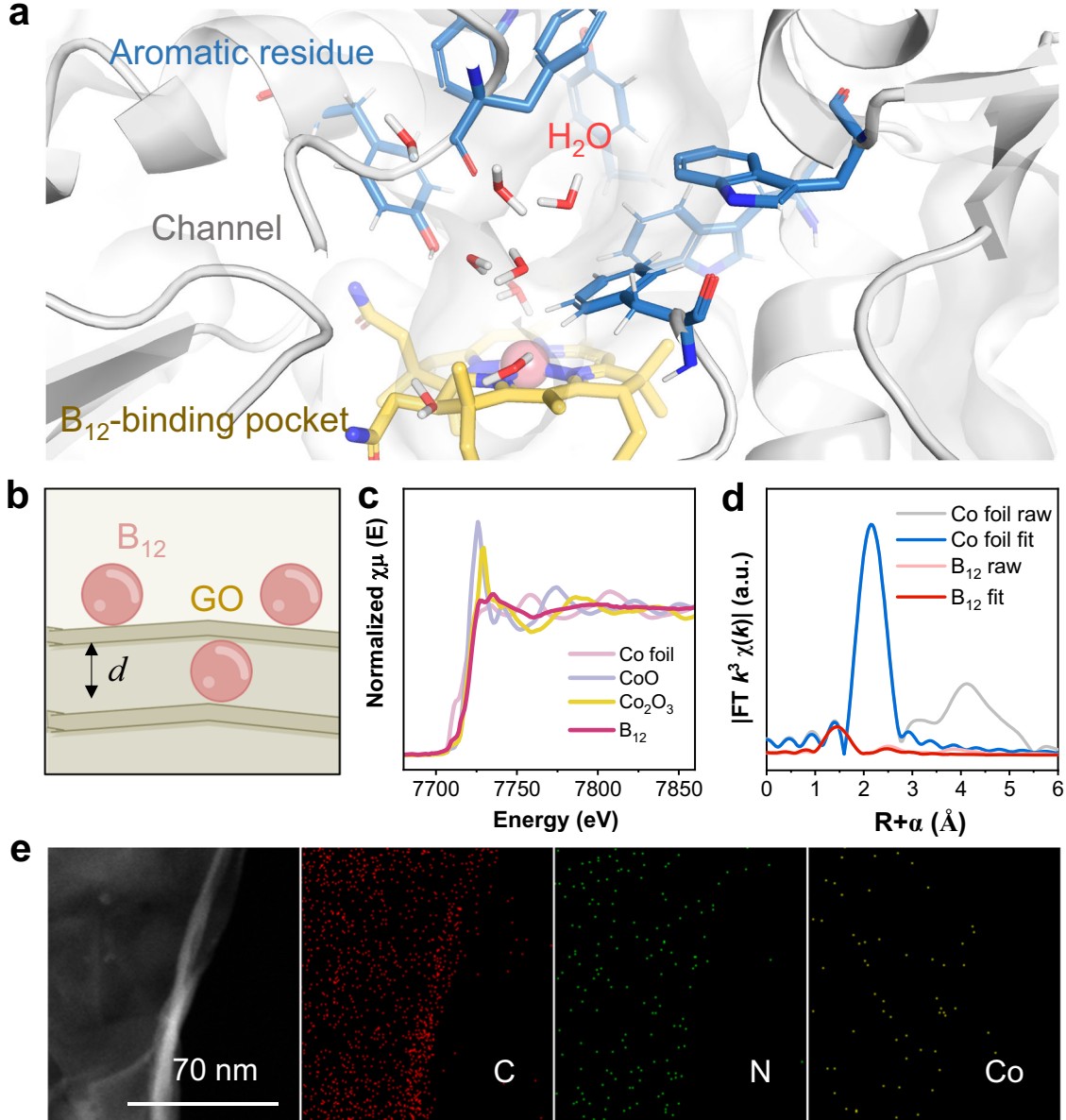

**Fig. 1 | Development of the binding pocket-like electrodes. a** $B_{12}$-dependent reductive dehalogenase (Protein Data Bank ID: 4UR0) and the binding pocket wrapped with aromatic residues, giving a channel accessible to the $B_{12}$ active site for the transfer of water and halogenated substrates. **b** Schematic illustration of the assembled GO-$B_{12}$-GO electrode. **c** XANES spectra and **d** Fourier transform (FT) at the Co K-edge of Co foil and $B_{12}$. **e** HAADF-STEM image of the GO-$B_{12}$-GO electrode with EELS element mapping.

(Fig. 2c), corresponding to an interlayer distance of 0.880 nm[33]. At a peak of 8.127°, the intercalation of $B_{12}$ was observed in the GO-$B_{12}$-GO electrode with a larger interlayer distance of 1.088 nm (Fig. 2c). Moreover, small-angle X-ray scattering (SAXS) patterns show a well-defined isotropic scattering at an azimuthal angle of 90°, indicating the enhanced alignment of GO sheets and the formation of a layer-by-layer structure in GO-$B_{12}$-GO after electrochemical treatment (Supplementary Fig. 12).

To evaluate the contribution of $B_{12}$ to the electrochemical reduction of TCA, the loading of $B_{12}$ was assessed by UV–vis absorption spectroscopy. Determined by the absorption peak at 360 nm (Fig. 2d and Supplementary Fig. 13), a loading of 0.0576 mg cm$^{-2}$ was achieved for GO-$B_{12}$-GO while a lower content was obtained for GO-$B_{12}$ (0.0365 mg cm$^{-2}$) or $B_{12}$ alone (0.0363 mg cm$^{-2}$), suggesting that the sufficient loading of $B_{12}$ for GO-$B_{12}$-GO was ascribed to its intercalation into the inner space (Fig. 2e). In addition, the electrochemical activation of GO-$B_{12}$-GO electrode resulted in an increased current density

with CV profiles (Fig. 2f and Supplementary Fig. 14) due to the deoxygenation of GO sheets and restoration of π-conjugated system[34,35]. The cathodic peaks for the Co(III)/Co(II) and Co(II)/Co(I) redox couples of GO-$B_{12}$-GO were separately observed at 0.61 and −0.26 V (vs. NHE) at a scan rate of 40 mV s$^{-1}$ (Supplementary Fig. 15). As the scan rate increased from 40 to 200 mV s$^{-1}$, the cathodic peak potential shifted negatively due to irreversible polarization. In the region without redox peaks, the current density at 0.2 V (vs. NHE) was used to estimate electrochemically active surface area (ECSA). Although the irreversible polarization at high scan rates may cause a residual current, a significant linear relationship between the current and scan rate was observed (Fig. 2g), indicating that the disturbance from residual current could be neglected[36,37]. Since the linear slope is proportional to surface area, the estimated ECSA of GO-$B_{12}$-GO electrode was almost 3.8 times larger than that of GO-$B_{12}$ and 27 times higher than that of $B_{12}$ alone. As such, the inner surface was accessible at a large interlayer distance of GO-$B_{12}$-GO but was limited in the case of GO-$B_{12}$.

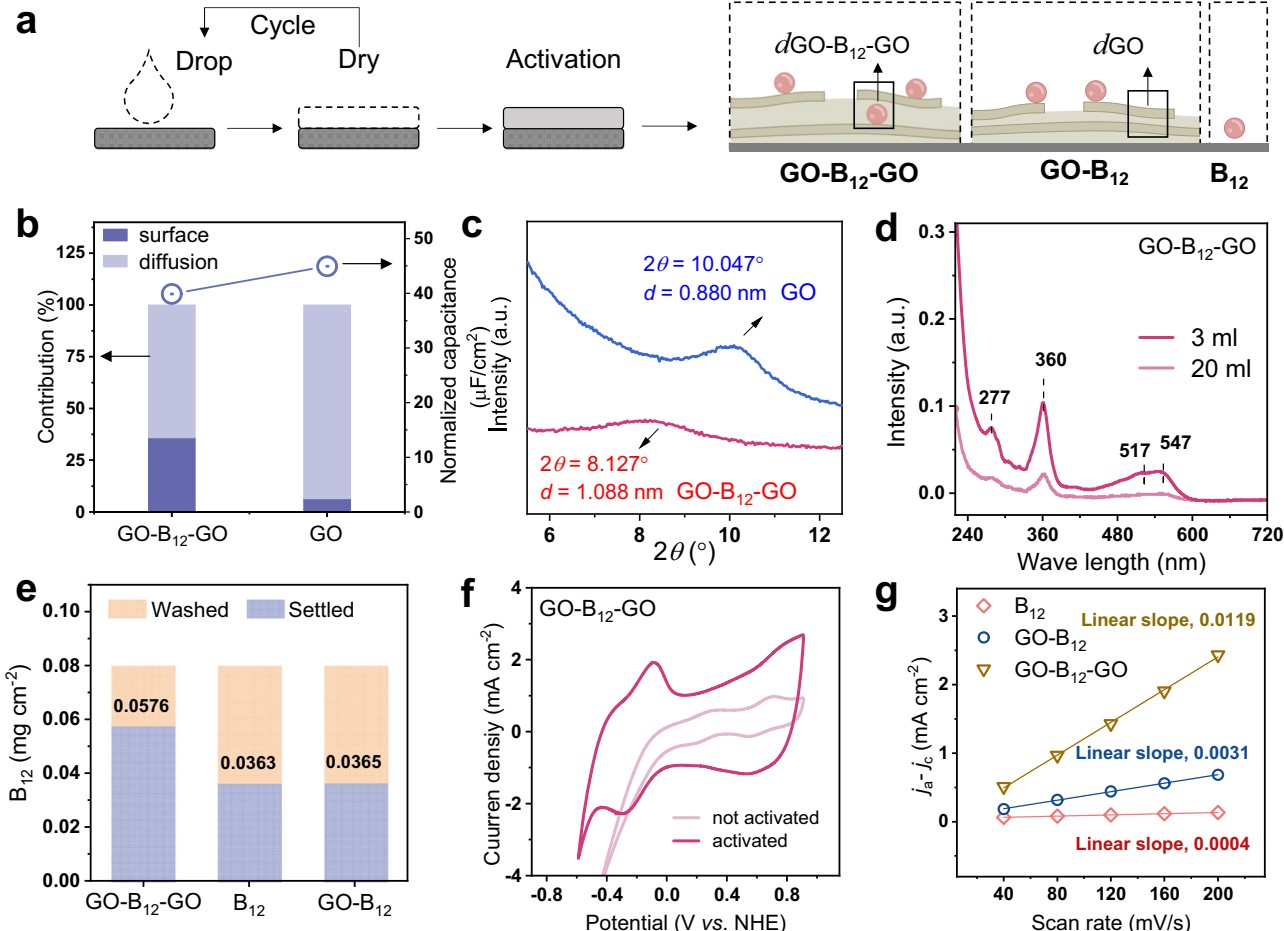

**Fig. 2 | Electrochemical activation and characterizations. a** Schematic illustration of the preparation of electrodes, with steps involving drop−dry cycling and electrochemical activation. **b** Normalized capacitance (right axis) and contribution proportion of the diffusion process and the surface process (left axis) for GO-B$_{12}$-GO and GO electrodes. **c** XRD profiles for analyzing interlayer distance, $d_{\text{GO-B12-GO}}$, and $d_{\text{GO}}$. **d** UV–vis absorption spectra for quantifying B$_{12}$ content: the diffused B$_{12}$ (20-mL electrolyte) and the loosely bound B$_{12}$ removed from the activated GO-B$_{12}$-GO electrode by ultrasonication (3-mL water). **e** B$_{12}$ content at different electrodes. **f** CV plots of the GO-B$_{12}$-GO electrode after activation. **g** Linear fitting of capacitive current *vs.* scan rates to estimate the ECSA of different B$_{12}$-based electrodes. The current densities at 0.2 V *vs.* NHE in the non-Faradaic region were used for $j_a$–$j_c$.

---

Additionally, in situ attenuated total reflectance Fourier transform infrared spectroscopy (ATR-FTIR) measurements were further conducted, revealing that the intercalated B$_{12}$ in GO-B$_{12}$-GO plays a critical role in facilitating the electrochemical reduction (see details in Supplementary Note 3, Supplementary Fig. 16−18).

**Electrochemical reduction of TCA in water**

The electrochemical catalytic performance of GO-B$_{12}$-GO electrode was evaluated using a conventional three-electrode system. For the TCA reduction, the GO electrode exhibited a low conversion efficiency of 17% (Fig. 3a). In contrast, the B$_{12}$-based electrodes presented higher conversion efficiencies, among which GO-B$_{12}$-GO achieved the highest conversion efficiency of TCA (86%). Additionally, the mass balance of chlorine was determined in the electrochemical reaction process (Fig. 3c). The corresponding apparent rate constant of TCA conversion ($k_1$) was 0.64 h$^{-1}$ for GO-B$_{12}$-GO, which was slightly higher than those of GO-B$_{12}$ and B$_{12}$ (Fig. 3d). However, for the detected intermediate of sodium dichloroacetate (DCA), the apparent rate constant ($k_2$) of GO-B$_{12}$-GO was 1.49 h$^{-1}$, 7.8 and 5.7 times higher than those of GO-B$_{12}$ and B$_{12}$, respectively (Fig. 3e). Moreover, the higher activity of DCA conversion by the GO-B$_{12}$-GO electrodes was confirmed with the released chlorine ions (Fig. 3b). The product of the DCA reduction was identified as sodium monochloroacetate (MCA), showing an overall reaction of TCA→DCA→MCA. Further analysis was performed to understand the high performance of GO-B$_{12}$-GO electrode. The catalytic activity of B$_{12}$ was assumed to be equal between GO-B$_{12}$ and GO-B$_{12}$-GO electrodes. Based on the activity toward DCA conversion of B$_{12}$-GO electrode, the theoretically calculated number of B$_{12}$ was $1.54 \times 10^{-7}$ mol cm$^{-2}$ required for the experimental performance of GO-B$_{12}$-GO electrode, far larger than the observed loading of B$_{12}$ ($4.25 \times 10^{-8}$ mol cm$^{-2}$). This indicates that the intercalated B$_{12}$ in the GO-B$_{12}$-GO electrode possesses a dramatically higher activity than the surface-adsorbed B$_{12}$, as illustrated in Fig. 3g–i. Moreover, the electrochemical dechlorination performance and the structural stability of GO-B$_{12}$-GO can be maintained after 15-hour electrolysis (Supplementary Figs. 19 and 20).

The information gleaned above demonstrates that the configuration of GO-B$_{12}$-GO electrode played a vital role in the sodium chloroacetate conversion activity. As such, we anticipate that other cobalt-centered molecules within the layered nanostructure may function similarly in electrochemical catalysis. To examine the applicability of such a sandwich-like electrode configuration, cobalt phthalocyanine (CoPc) was used as an alternative cobalt-centered catalyst (see details in Supplementary Note 4, Supplementary Figs. 21−24). For the GO-CoPc-GO electrode, the TCA conversion efficiency after 3 h of electrolysis reached 80%. In sharp contrast, the GO-CoPc electrode only displayed a substantially lower conversion

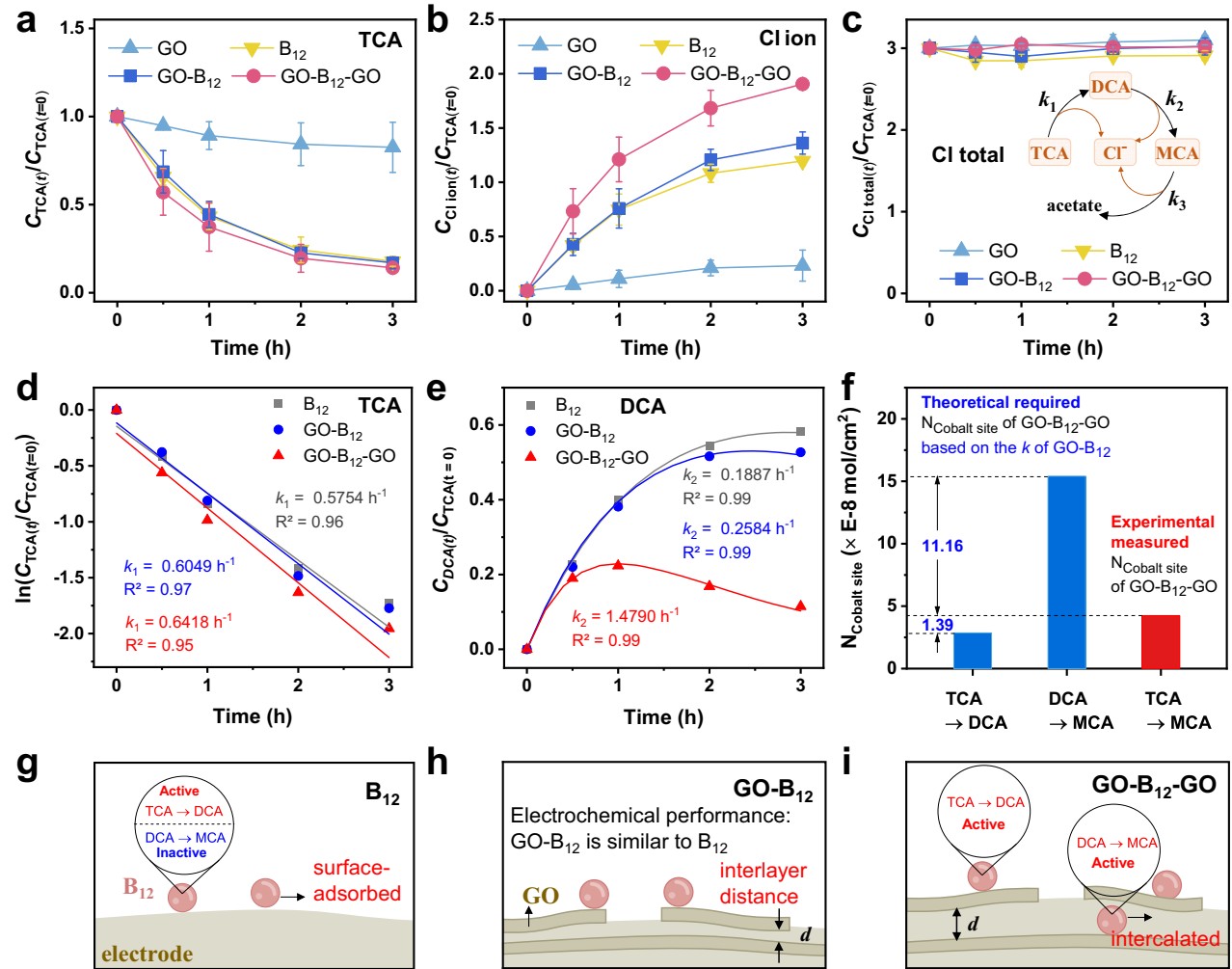

**Fig. 3 | Electrochemical reductive dechlorination performance.**
**a**, **b** Concentration ratios of TCA and Cl⁻ ions during the 3-h electrolysis at GO, $B_{12}$, GO-$B_{12}$, and GO-$B_{12}$-GO electrodes. Reaction conditions: −0.29 V *vs.* NHE, pH = 7.0, carbon fiber paper working electrode. **c** Mass balance of chlorine elements, including TCA, DCA, MCA, and the released Cl⁻ ion. Error bars indicate standard deviations obtained from two independent measurements. **d**, **e** Apparent rate constants of TCA conversion ($k_1$) and DCA conversion ($k_2$). **f** Number of cobalt sites on GO-$B_{12}$-GO electrode. The blue column is the $N_{cobalt\ site}$ required to achieve the experimental performance of GO-$B_{12}$-GO via calculations based on $k_1$ and $k_2$ of GO-$B_{12}$. The red column is the loaded $N_{cobalt\ site}$ quantified by UV–vis absorption spectroscopy. **g**–**i** Schematic illustrations of the electrochemical performance of $B_{12}$, GO-$B_{12}$, and GO-$B_{12}$-GO electrodes.

efficiency of 20%. This verifies that such a sandwich-like electrode configuration improved the electrochemical reduction of TCA. However, it is worth pointing out that in contrast to GO-$B_{12}$-GO, the use of GO-CoPc-GO led to DCA accumulation (TCA→DCA) due to the limited reductive conversion of DCA to MCA (Supplementary Figs. 25 and 26). To probe the difference in the performances between GO-$B_{12}$-GO and GO-CoPc-GO, DFT calculations were conducted to analyze the effects of the CN and symmetry of $B_{12}$ and CoPc on the cleavage of C−Cl bonds. In contrast to the symmetric CoPc, an asymmetric Co-corrin was able to activate the C−Cl bond with a larger bond length of 1.843 Å (Supplementary Fig. 27). Moreover, the axial ligand (5,6-dimethyl-benzimidazole) formed a five-coordinated Co site of $B_{12}$, resulting in the fully activated C−Cl bond with 1.925 Å. Partial density of states (PDOS) analysis shows an energy overlap between the *p* orbitals of DCA and the *d* orbitals of $B_{12}$. However, a mismatch of the energy levels was observed between the molecular orbitals of DCA and CoPc (Supplementary Fig. 28). The large overlap populations reveal the interactions between the DCA ion and the Co site of the catalyst, which can be used as a descriptor for reduction catalysis[38]. For this reason, the Co site of $B_{12}$ exhibited a higher catalytic activity for the reductive conversion of DCA compared with the Co site of CoPc.

## Intercalation process and solvation structures

In the heterostructure of GO-$B_{12}$-GO, $B_{12}$ at the electrolyte−electrode interface was assumed to play a key role in electrocatalytic TCA conversion. With TCA as a molecular probe, an increased current density was observed in the TCA electrolyte compared with the blank electrolyte, indicating that the GO-$B_{12}$-GO electrode was catalytically active to reduce TCA. However, a decrease in current density occurred under the same conditions for the GO electrode, suggesting that GO alone was catalytically inactive to TCA (Fig. 4a). Upon adding $B_{12}$ to the electrolyte, $B_{12}$ could diffuse from the bulk electrolyte to the electrode driven by a concentration gradient, forming a second layer of $B_{12}$ at the surface. However, the reduction current density of TCA did not increase with the increasing number of $B_{12}$ (Supplementary Fig. 29), suggesting that the more surface-adsorbed $B_{12}$ cannot contribute to the current density. To further examine the role of $B_{12}$, nitrogen-doped graphene (GN) was also used to assemble the GN-$B_{12}$-GN electrode driven by vdW forces (see details in Supplementary Note 5, Supplementary Figs. 30–32). The electrocatalytic contribution of the GN-$B_{12}$-GN electrode was consistent with that of GO-$B_{12}$-GO (Fig. 4b).

To understand the roles of the intercalated $B_{12}$ and the corresponding interlayer space in the DCA conversion, MD simulations

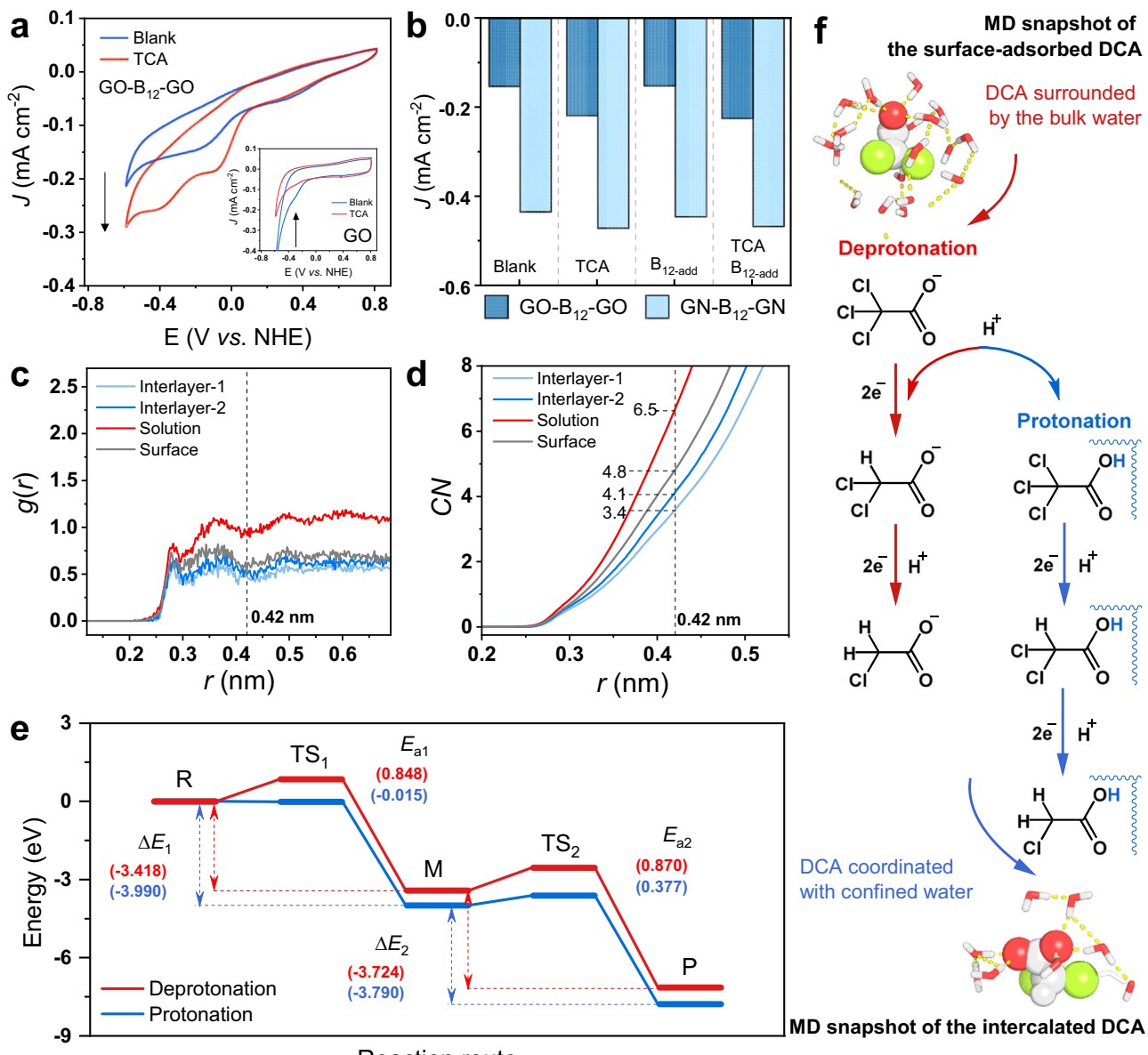

**Fig. 4 | Dechlorination of DCA in the interlayer space of binding pocket-like electrocatalyst. a** CV profiles of GO-B$_{12}$-GO electrode at a scan rate of 200 mV s$^{-1}$ (insets for GO alone). **b** Current density of GO-B$_{12}$-GO at −0.29 V vs. NHE (pH = 7.0) ("B$_{12\text{-add}}$" for more B$_{12}$ added to the electrolyte). The GN-B$_{12}$-GN electrode was used as an alternative B$_{12}$-intercalated electrode. A glassy carbon electrode was used as the working electrode. **c** Radial distribution function $g(r)$ for the DCA−O$_w$ distance between DCA and the O atom of water. **d** Coordination number (CN) of water. **e** Energy profiles for the reductive dechlorination pathway in the (de)protonation states ("R" for TCA, "M" for DCA, "P" for MCA, "$\Delta E$" for reaction energy change, and "$E_a$" for reaction barrier). **f** Schematic illustration of the proposed protonation states. For the model of DCA, light grey, carbon; red, oxygen; green, chlorine.

were conducted. As illustrated in Supplementary Movies 1 and 2, four DCAs with Na$^+$ counter ions were initially located in the bulk solution, far away from the GO-B$_{12}$-GO and GO-B$_{12}$ electrodes. Subsequently, DCA started to diffuse and reached an equilibrium state, where two DCA molecules were intercalated into the space between the GO sheets, with one adsorbed on the GO surface and the other still diffusing randomly in aqueous solution (Supplementary Fig. 33). The radial distribution function $g(r)$ shows the first narrow peak followed by two wide peaks, giving the distance between DCA and water molecules (DCA−O$_W$) of 0.28, 0.36, and 0.49 nm, respectively (Fig. 4c). Accordingly, the average CN of DCA shows an order from high to low as follows: solvated in solution (CN = 6.5), adsorbed on the surface (CN = 4.8), and intercalated within the inner space (CN = 4.1, 3.4). The values of CN were calculated within the second shell of solvation at a DCA-O$_W$ distance of 0.42 nm, which was

confirmed by doubling the run time to 50 ns (Fig. 4d and Supplementary Fig. 34). Compared with the bulk water, a less tight structure of water was found around the intercalated DCA (Supplementary Fig. 35), which was driven by the electrostatic and hydrogen bonding interactions of GO and B$_{12}$ similarly to the local environment within the enzyme[22]. Moreover, DFT calculations were performed for the reduction pathways of DCA in the interlayer space. In the bulk solution, DCA was deprotonated due to the experimental pKa of 1.26[39], giving an energy barrier of 0.870 eV for the dechlorination reaction (DCA → MCA). In comparison, DCA in the interlayer space was partially solvated, providing a chance to alter the protonation state, which would facilitate dechlorination via a lower barrier (0.377 eV, Fig. 4e). As such, the dechlorination reaction pathway can be regulated by the solvation structure around the chlorinated organic pollutant (Fig. 4f).

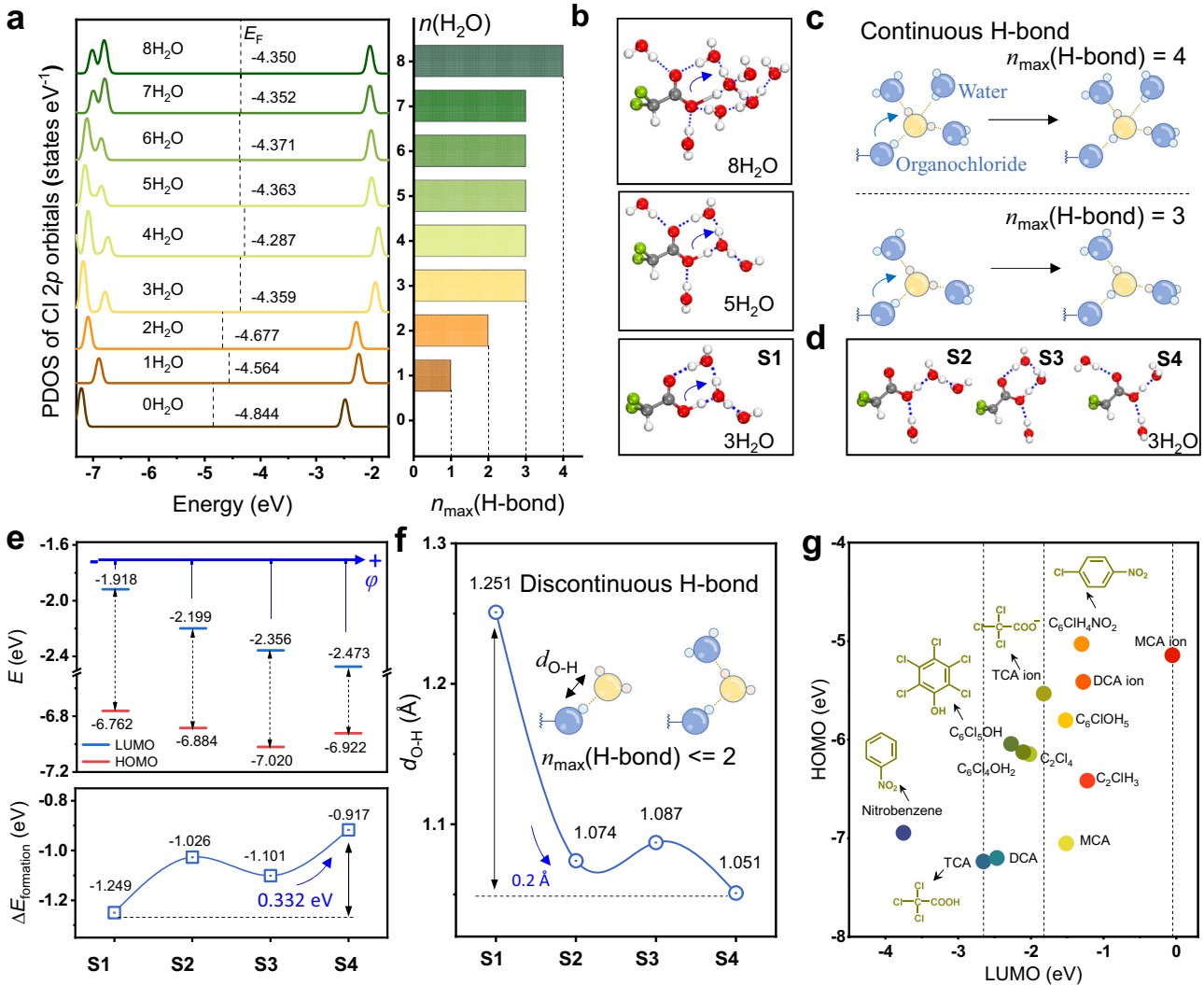

**Fig. 5 | Hydrogen-bond network in nanospace for promoting reductive dechlorination. a** PDOS for Cl $2p$ orbitals of DCA with the number of coordinated water varied from 0 to 8. Each state has been relaxed to the lowest energy configuration, and $n_{max}$(H-bond) is the maximum number of hydrogen bonds that a water molecule can form. **b** Proton transfer triggered by the continuous hydrogen-bond network of the relaxed DCA·$n$H$_2$O ($n$ = 3, 5, 8) with **c** the $n_{max}$(H-bond) of 3 or 4. **d** Other configurations of DCA·3H$_2$O (S2 to S4) in addition to the lowest energy configuration (S1). **e** The frontier molecular orbitals relative to the vacuum level ($E$), and the formation energy for the hydration process of DCA·3H$_2$O ($\Delta E_{formation}$) at the four configurations. **f** Protonation state of DCA in the discontinuous hydrogen-bond network with $n_{max}$(H-bond) of 1 or 2 ("$d_{O-H}$" for the O-H distance of DCA). **g** LUMO levels of various chlorinated organic pollutants.

## Mechanism of hydrogen-bond network for promoting dechlorination

The coordination structure of water around DCA was in dynamic equilibrium, and consequently, the conformation space would be relatively large. To simulate the hydrogen-bonding network in bulk water, DCA·$n$H$_2$O was relaxed to the lowest energies without structural confinement, resulting in a trend of increasing band gap of DCA with increasing water number, as shown in the PDOS plot (Fig. 5a). The MD simulations of DCA in bulk (CN = 6.5, within 0.42 nm) show that the conductive band minimum (CBM) of DCA was at −2.0 eV ($n$ = 5–8). In contrast, DCA in the gas phase ($n$ = 0) showed the lowest Fermi level of −4.844 eV with a CBM of −2.467 eV, which favors accepting electrons from the external cathode. In the partially solvated states ($n$ = 1–4), the CBM was in the range from −2.409 to −1.937 eV. Orbital analysis shows that the lowest unoccupied molecular orbitals (LUMO) were located on the skeleton of the C−Cl bond for DCA (Supplementary Fig. 36). As a result, DCA with a lower LUMO would undergo a more favored reduction of the C−Cl bond. Moreover, the continuity of the hydrogen-bond network was demonstrated by the maximum number of hydrogen bonds per water molecule formed. In addition to the lowest energy

structures, the increased energy by the steric strain of the discontinuous hydrogen-bond network could regulate the desolvation structure of DCA. To investigate the hindrance effect of the electrode configuration on the solvation structure and hydrogen-bond network, $n$ = 3 was taken as an example with four possible states (S1 to S4, Fig. 5b, d). With the higher formation energy (S2 to S4, Fig. 5d), DCA·3H$_2$O became less stabilized. In the meantime, the LUMO energy level showed a negative shift from −1.918 to −2.473 eV (Fig. 5e), suggesting that the acceptance of electrons was more favored. Moreover, the distance of O−H showed a decrease from 1.251 to 1.051 Å (Fig. 5f), resulting in a state change from deprotonation (COO$^-$) to protonation (COOH). Deprotonated DCA (S1) showed sequential proton diffusion via the Grotthuss mechanism known in bulk (Fig. 5b), depending on the interactions between the hydronium ion and water[40–42]. In contrast, the protonated DCA (S2 to S4) showed a well-preserved O−H bond due to the discrete hydrogen-bonding network (Fig. 5d), which would facilitate the reductive dechlorination with lower energy barriers (Fig. 4e).

As mentioned above, an improved electrochemical catalytic reduction was achieved by the layered GO-B$_{12}$-GO electrode via the

steric effect on the protonation state and the required overpotential. To achieve the reduction of more small substrates, the reactivity for the target substrate is assessed by the relationship of their molecular orbitals and electrochemical potentials (Fig. 5g). For chloroacetate, the order of the LUMO energy level was found to be MCA>DCA>TCA, which agrees with the order of the experimental conversion efficiency. Consistent with this rule, nitrobenzene, a known substrate undergoing electrochemical reduction more easily than chloroacetates, exhibits a substantially lower LUMO energy level. Other chlorinated organic substrates, such as chlorophenol and chloroethane, are also examined, providing a reference foundation for the difficulty of electrochemical reduction. In addition, the size matching of the small substrates and the interlayer space may play important roles in the desolvation of substrates and further the transformation efficiency.

## Discussion

This work demonstrates that the binding pocket-like electrode based on molecular catalysts ($B_{12}$) and 2D materials (GO) is capable of providing inner space to boost the electrochemical reduction of C−Cl bonds. The desolvation of the intercalated chlorinated organic substrates can alter the protonation state through discontinuous hydrogen bonding network, minimize the required overpotential by favoring electron transfer, and thereby accelerate the reduction reactions. In addition to the surface reaction, the inner space in electrodes can be used to tune the catalytic reduction performance. Given a large variety of available molecular catalysts, the integration of precise active sites with dynamic inner space is an applicable strategy for various electrochemical catalytic reactions. By taking advantage of the continuous advances in 2D layered nanomaterials and 3D porous nanomaterials, this work will provide a foundation for establishing regulation approaches for Van der Waals heterostructures and constructing high-performance electrocatalysts for water treatment.

## Methods

### Preparation of binding pocket-like electrode

The sandwich-like electrode based on graphene oxide and molecular catalyst was prepared by drop−dry cycles and electrochemical pretreatment. A mixture of GO (5 mg mL$^{-1}$, 0.2 mL) and molecular catalyst (4 mg mL$^{-1}$, 0.2 mL) was dissolved in a solution of Nafion (5 wt%, 0.05 mL), DI water (0.2 mL) and ethanol (0.4 mL for $B_{12}$) or dimethylformamide (0.4 mL for CoPc). Then, the obtained catalyst ink was sonicated for 1 h. For the GO-$B_{12}$-GO electrode, 40 μL of the prepared solution was dropped onto carbon fiber paper (1.0 cm × 2.0 cm) and dried under an infrared lamp. A total of 200 μL of the catalyst ink was loaded in 5 cycles, followed by CV measurement at a scan rate of 200 mV s$^{-1}$. The fast cyclic test was efficient for removing the excess molecular catalysts and forming a well-layered structure by deoxygenation and restacking of GO sheets. The sandwich-like electrode was repeatedly washed in water and dried in air for further use. Other electrodes were prepared with a similar process. For the GO-only electrode, a GO ink was obtained by mixing GO (5 mg mL$^{-1}$, 0.2 mL) with DI water (0.4 mL) and ethanol (0.4 mL) under sonication. For the $B_{12}$-only electrode, a $B_{12}$ ink was prepared by dissolving $B_{12}$ (4 mg mL$^{-1}$, 0.2 mL) in a solution of Nafion (5 wt%, 0.05 mL), DI water (0.4 mL), and ethanol (0.4 mL). For the GO-$B_{12}$ electrode, the as-prepared GO electrode was then loaded with the $B_{12}$ ink under the same conditions.

### Characterizations

GIXRD measurements were performed on a X'PertPro MPD (Panalytical) with Cu Kα radiation (λ = 0.15406 nm). The morphology of the prepared materials was observed with TEM (H-7650, Hitachi Co., Japan). High-resolution TEM, HAADF-STEM, and EELS mapping were carried out by a JEM-ARM200F TEM/STEM (JEOL Co., Japan). The defect degree of carbon was characterized by Raman spectroscopy

(LABRAM HR EVO, Horiba Co, Japan). XAFS (Co K-edge) were obtained at Beamline 11B from the Shanghai Synchrotron Radiation Facility, China. Co K-edge XANES data were collected in fluorescence mode and analyzed using the ATHENA module implemented in the IFEFFIT package[43]. A cubic spline function was used to fit the background above the absorption edge. To amplify the EXAFS oscillations in the mid-$k$ region, $k^2$ weighting was applied, followed by a Fourier transform process for converting the data to a radial distribution ($R$) space with a $k$ range of 2.5–12 Å$^{-1}$ at the Co K edge. High-resolution X-ray absorption spectra were collected at the Catalysis and Surface Science Endstation at the BL11U Beamline in the National Synchrotron Radiation Laboratory (NSRL) in Hefei, China. SRPES measurements were conducted at the photoemission end-station at the BL10B Beamline in the NSRL in Hefei, China.

### Electrochemical measurements

All electrochemical measurements were conducted using a potentiostat (CHI 750E, Chenhua Co., China) in a three-electrode configuration. A platinum wire 0.5 mm in diameter was used as the counter electrode. Potentials were measured against an Ag/AgCl reference electrode. The supporting electrolyte utilized for all experiments was prepared by using 0.067 M Na$_2$HPO$_4$ and KH$_2$PO$_4$ (PBS) solutions. For CV measurements, a glassy carbon electrode (GCE) with a diameter of 3 mm was used as the working electrode. Each GCE was loaded with an aliquot of 6 μl of the catalyst ink and dried in air, giving a loading of 0.34 mg/cm$^2$. For constant-potential electrolysis, an H-type cell was used with a cathode cell (100 mL) and anode cell (100 mL) separated by a proton exchange membrane (Nafion−117, Du Pont Co., USA). Carbon fiber paper (thickness 0.28 mm, TGP-H-090, Toray Co., Japan) was used as the working electrode. In the electrolysis, the electrolyte was agitated with a stirring bar at a rate of 350 rpm. The concentrations of chloroacetate ions (tri-, di- and mono-) and chloride ions were measured by ion chromatography (ICS-1000, Dionex Co., CA). The mobile phase was 15 mM KOH with a flow rate of 1.0 mL min$^{-1}$. SAXS data were collected on a point collimated Anton Paar's SAXS point 2.0 system equipped with a microfocus X-ray source (Cu Kα radiation, wavelength 1.5418 Å), and an Eiger R 1 M Tilt detector with a pixel size of 75 × 75 μm$^2$. In situ ATR-FTIR was performed on Thermo-Fisher Nicolet iN10 with liquid N$_2$-cooled HgCdTe (MCT) detector. A custom-made three-electrode electrochemical single-cell was used for electrochemical tests. Ge tip with a fixed angle (27°) was used for collecting FTIR spectra. Pt-wire and Ag/AgCl electrodes were used as counter and reference electrodes, respectively. PBS (0.067 M, pH = 7) was used as an electrolyte. For ATR-FTIR measurement, 16 scans were collected with a spectral resolution of 4 cm$^{-1}$ to collect data.

### Reporting summary

Further information on research design is available in the Nature Portfolio Reporting Summary linked to this article.

## Data availability

All data that support the findings in this paper are available within the article and its Supplementary Information files. Source data are provided in this paper.

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

## Acknowledgements

The authors wish to thank the National Natural Science Foundation of China (52022093, 51978637, 51821006, 52192684, and 22232003), the China Postdoctoral Science Foundation (2022M723036), Xiaomi Young Talents Program, USTC Research Funds of the Double First-Class Initiative (YD3530002001), Youth Innovation Promotion Association, and the Fundamental Research Funds for the Central Universities (WK2400000004) for the partial support of this work. We thank the photoemission endstations BL1W1B in Beijing Synchrotron Radiation Facility (BSRF), BL14W1 in Shanghai Synchrotron Radiation Facility

(SSRF), and BL10B and BL11U in National Synchrotron Radiation Laboratory (NSRL) in Hefei for the help in characterizations. The numerical calculations in this paper were performed on the supercomputing system in the Supercomputing Center of the University of Science and Technology of China.

## Author contributions

Y.M., J.J.C., H.Q.Y., and Y.X. conceived and designed the research; Y.M. performed the DFT calculations and the electrochemical experiments; S.C.M. and X.Q.P. conducted the synthesis and characterization. J.J.C., H.Q.Y., and Y.X. contributed to the planning and coordination of the project. Y.M., J.J.C., H.Q.Y., and Y.X. co-wrote and edited the manuscript. All the authors contributed to the discussion of the results and the manuscript.

## Competing interests

The authors declare no competing interests.
