## [Peer Review File · Nature Communications]

Mimicking reductive dehalogenases for efficient electrocatalytic water dechlorinationREVIEWER COMMENTS

Reviewer #1 (Remarks to the Author):

The authors report a timely and robust demonstration of enzyme activity in a layered GO assembly that could potentially be used to reduce contaminants from wastewater. The expts and models are generally well performed and the data supports the conclusion.

On the exptl side, the work could be improved by discussing the limits of in situ characterisation of the protein structure (fold) in the GO structure as in operando monitoring of the protein structure would be crucial for deployment as sensors (as well as inferred "protein health" from the activity measurements).

On the modelling side, consideration of alternative dispersion scheme in DFT, eg D3, would be useful to compare with TS given the important of vdW effects in the assembly. Similarly on the MD side, the model choice for water is somewhat arbitrary and the water confinement is a crucial feature, eg, CHARMM36m water (doi: 10.1038/nmeth.4067) or other water model could give more insights. Also MD simulations are short given the modest model size; are all timelines, not just the interlayer spacing, converged in the few nanoseconds? Consider doubling the run time and/or performing duplicate runs from different starting configurations to enhance sampling.

Reviewer #2 (Remarks to the Author):

This manuscript developed an enzyme-like GO-B12-GO heterostructure electrocatalysts for selective electroreduction of sodium trichloroacetate(TCA). The catalytic center (vitamin B12) of dehalogenase in organohalide-respiring bacteria was chosen as the molecular catalyst to construct the GO-B12-GO heterostructure. The results showed that the electrocatalyst exhibits excellent dechlorination performance, which enhances reduction of intermediate dichloroacetic acid by 7.8 folds against that without sandwich configuration and can selectively generate monochloro-groups from trichloro-groups. Molecular simulations as well as DFT are used to reveal the mechanism. Overall, the design strategy in this work is novel, the experiments are performed in high quality, and the manuscript is written well. However, there are a few concerns as given below.

- 1.As the authors mentioned in the manuscript, all the purified reductive dehalogenases contain cobalamin cofactor (i.e., vitamin B12 and its derivatives), it that the synthesized GO-B12-GO heterostructure a kind of combination of GO and enzyme instead of mimic enzyme?
- 2.The interface behavior of the synthesized GO-B12-GO heterostructure is unclear which is needed to be more investigated.
- 3.Did the authors analyze the effects of different loading amounts of B12 on the electrochemical catalytic performance?
- 4.The reason why the intercalated B12 in the GO-B12-GO electrode possesses a dramatically higher activity than the surface-adsorbed B12 is unclear just based DFT or MD simulations. Some in situ characterizations such as in situ FTIR or in situ Raman are recommended to further analysis the mechanism of the reaction.
- 5.In general, the catalyst in this work is neither a homogeneous nor a heterogeneous catalyst, but a coupling of both. It cannot provide the possibility of catalyst recovery with unlost catalytic activity, nor guarantee the long-term effectiveness of biological catalysts due to the environment-sensitive enzyme (CbFDH) is directly exposed to the reaction solution.
- 6.As the author emphasized that the sandwich-like configuration of GO-B12-GO heterostructure played an essential role in enhancing the performance, however, there is no obvious characterization result to prove the sandwich-like configuration.

Reviewer #3 (Remarks to the Author):

In this study, the authors proposed and developed the GO-B12-GO electrode for toward water dechlorination by mimicking the binding pocket configuration and catalytic center of reductive dehalogenases. The assembled heterostructure electrode by sandwiching a molecular catalyst into the interlayers of two-dimensional graphene oxide exhibits excellent dechlorination performance, which enhances reduction of intermediate dichloroacetic acid by 7.8 folds against that without sandwich configuration. Furthermore, the authors made efforts to clarify why and how the GO-B12-GO works well for water dechlorination by means of DFT calculations and molecular simulations. The calculation results suggest that the sandwiched inner space plays an essential role in tuning solvation shell, altering protonation state and facilitating carbon-chlorine bond cleavage. Seemingly, the electrocatalytic performance of the catalyst the authors developed was good and the mechanism was proposed was interesting to researchers in the fields of science and technologies. However, taking a close look at the details of the manuscript, I wanted to more clearly understand the followings, which are unfortunately not described in the manuscript.

I also have the following points that need to be better proven in the manuscript:

- 1 : Whether the electronic structure of Co center in B12 in the sandwich structure is the same as that of B12 alone. This comparison is important for understanding the catalytic mechanism.
- 2 : In addition to GIXRD, more experimental data are needed to support the GO-B12-GO structure.
- 3 : Why do you think that the difference in performance of B12 and CoPc comes from the difference in the interaction of DCA with B12 and CoPc, but is related to the difference in spatial structure between GO-CoPc-GO and GO-B12-GO?
- 5 : What is the interlayer distance of the GO-CoPc-GO electrode?
- 4 : In Supplementary Fig. 13, Two CV curves correspond to without TCA and with TCA, in the figure, what does means of "arrow and adding B12 in the bulk"?
- 5 : In addition to the role of solvent, is the adsorption of TCA molecules and reaction intermediates on B12 also affected by space? How does the change in adsorption affect the reaction process?
- 6 : It would be better for the authors to show the stability of the structure and performance of catalyst or electrode.

Response to Reviewer 1's Comments

The authors report a timely and robust demonstration of enzyme activity in a layered GO assembly that could potentially be used to reduce contaminants from wastewater. The expts and models are generally well performed and the data supports the conclusion.

We really appreciate the referee's highly positive evaluation of our work, and are grateful to the referee for his/her comments and suggestions to help us further improve the quality of our manuscript. We have made all the revisions as suggested by the referee.

On the exptl side, the work could be improved by discussing the limits of *in situ* characterisation of the protein structure (fold) in the GO structure as in operando monitoring of the protein structure would be crucial for deployment as sensors (as well as inferred "protein health" from the activity measurements).

We thank the referee for his/her insightful suggestion. According to the referee's suggestion, both *ex situ* and *in situ* additional measurements were performed to investigate the structural properties of B₁₂ in various electrode configurations. The corresponding descriptions and supplementary figures have been supplemented in the manuscript and the SI (Supplementary Figures 16-18), respectively.

For the *ex situ* characterizations, the Raman spectra of the B₁₂ and GO-B₁₂ electrode (Figure R1a) exhibit a prominent band at 1492 cm⁻¹, indicating the presence of vibrational modes associated with C=C stretching, C=N stretching and C-H bending within the corrin rings of B₁₂ molecules. The intercalation of B₁₂ into the GO-B₁₂-GO electrode results in a blue shift of the band (1502 cm⁻¹), which can be attributed to the π -stacking of the intercalated corrin ring fragment onto the GO layer (Sabater et al., *ACS Catal.* 2014, 4, 2038–2047). Additionally, two bands at 1341 and 1586 cm⁻¹ indicate the disorder-*sp*³ and graphite-*sp*² carbon components within the structures of GO, GO-B₁₂ and GO-B₁₂-GO. Similar comparison of the Fourier-transform infrared (FTIR) spectra (Figure R1b) show distinct peaks for B₁₂, including the cyanide stretching frequency at 2133 cm⁻¹, amide side chains at 1662 cm⁻¹, and breathing modes of the corrin ring at 1575 and 1550 cm⁻¹ (Chang et al., *Energy Environ. Sci.* 2012, 5, 5305-5314). In the fingerprint region, the GO-B₁₂ electrode exhibits the presence of two additional peaks at 1218 and 1145 cm⁻¹ compared to GO. Meanwhile, a noticeable red shift of the band, specifically to 1203 cm⁻¹, is observed in the case of the GO-B₁₂-GO heterostructure. Based on the above analysis of corrin ring and side chains, B₁₂ molecules in GO-B₁₂-GO retain the majority of their characteristic peaks and undergo additional interactions with GO nanosheets.

Figure R1. Spectroscopic Characterization. (a) Raman spectra of electrodes with various components, including GO, B₁₂, GO-B₁₂ and GO-B₁₂-GO, employing an excitation line of 532 nm. (b) FTIR spectroscopic measurements conducted on both the top and bottom surfaces of the corresponding layered electrodes.

To investigate the electronic states of B₁₂ molecules for further insights into the structure, the time-correlated single photon counting (TCSPC) characterizations were conducted on both the GO-B₁₂ and GO-B₁₂-GO electrodes by measuring the fluorescence lifetimes. As the instrument response function (IRF) captures the response of the whole architecture to a fast (sub-nanosecond) fluorescence emission, the corresponding full-width-at-half-max value of IRF is ~200 ps. The average lifetime (τ_{av}) is found to be ~4.345 ns for GO-B₁₂ and ~4.553 ns for GO-B₁₂-GO (Figure R2). The longer fluorescence lifetime of B₁₂ in GO-B₁₂-GO can be attributed to the long-range diffusion of few excitons in the extended π - π stacking channel (Zhou et al., *Nat. Catal.* 2023, <https://doi.org/10.1038/s41929-023-00972-x>). However, the lifetime of fast components is limited by the response function. This means that the charge transfer nature of B₁₂ was remained in both electrode configurations of GO-B₁₂ and GO-B₁₂-GO. Therefore, *ex situ* methods confirm the preservation of structural properties of B₁₂ within the GO layered structure.

Figure R2. TCSPC trajectories of GO-B₁₂ and GO-B₁₂-GO electrodes ($\lambda_{exc} = 375$ nm and

$\lambda_{em} = 420 \text{ nm}$).

For the *in situ* characterizations, attenuated total reflectance Fourier transform infrared spectroscopy (ATR-FTIR) measurements were further conducted to monitor the structural changes of the electrodes. GO-B₁₂-GO electrode was pretreated using electrochemical activation approach and dried in air before *in situ* experiments. In air state, the main FTIR bands are attributed to GO (Figure R1b and Figure R3), suggesting the low amount of B₁₂ used in the activated GO-B₁₂-GO electrode. Incorporation of water molecules in the layered GO-B₁₂-GO results in a blue-shift of bands, which reflects the strengthened interactions between water and the oxygen-containing groups of GO. Under electrochemical conditions, a significant reduction in the intensity of GO peaks ($\sim 1050 \text{ cm}^{-1}$) was observed as the applied potential from -0.2 V to -1.0 V vs. Ag/AgCl. Concurrently, the characteristic peaks of B₁₂ (1570, 1538, 1469, and 1446 cm^{-1}) were detected upon the reduction of GO components at -0.2 V vs. Ag/AgCl. Compared with the *ex situ* FTIR results, a red-shift of B₁₂ bands was found, which can be attributed to the interactions between corrin rings of B₁₂ and the reduced GO.

Figure R3. *In situ* ATR-FTIR spectra of GO-B₁₂-GO. Reaction conditions: -0.29 V vs. NHE, pH=7.0, carbon fiber paper working electrode.

On the modelling side, consideration of alternative dispersion scheme in DFT, eg D3, would be useful to compare with TS given the important of vdW effects in the assembly.

We thank the referee for his/her thoughtful suggestion. Following the referee's suggestion, the DFT-D3 dispersion method for the calculations was considered, and the results were compared with those from TS approach.

To account for the vdW interactions at the solution-electrode interface, DFT-D3 vdW method has been used to study the role of localized environment in the electric double layer (Tong et al., *Nat. Commun.* 2023, 14, 3397). Grimme's DFT-D3 corrected method for its effectiveness in describing the weak interactions, such as the long-range van der Waals and dispersive intermolecular interactions (Grimme et al., *J. Chem. Phys.* 2010, 132, 154104), was

employed in this study. Without any constraints on geometries, each state has been relaxed to the lowest energy configuration. DCA in the gas phase exhibits the lowest Fermi level of -4.821 eV with a conduction band maximum (CBM) of -2.438 eV (Figure R4). For the DCA $n\text{H}_2\text{O}$ assemblies in aqueous phase, the CBM exhibited a positive shift of 0.218 eV ($n = 1, 2$) and 0.572 eV ($n = 3\sim 8$). Compared with the values of CBM energy levels calculated using TS approach, a more significant energy shift was found for DCA $n\text{H}_2\text{O}$ using DFT-D3 method. Using the two versions of vdW exchange-correlation functional, we consistently conclude that the partial solvation structure of DCA favors the acceptance of electrons from the external cathode. In agreement with the referee's viewpoints, the vdW effects on the calculated CBM energy levels of different assemblies can be better described by DFT-D3 method (Brehm et al., *Nat. Mater.* 2020, 19, 43–48).

Figure R4. PDOS for Cl 2p orbitals of DCA with the number of coordinated H_2O varied from 0 to 8.

Similarly on the MD side, the model choice for water is somewhat arbitrary and the water confinement is a crucial feature, eg, CHARMM36m water (doi: 10.1038/nmeth.4067) or other water model could give more insights. Also MD simulations are short given the modest model size; are all timelines, not just the interlayer spacing, converged in the few nanoseconds? Consider doubling the run time and/or performing duplicate runs from different starting configurations to enhance sampling.

We thank the referee for his/her valuable suggestion. Following the referee's suggestion, MD simulations were performed using alternative water model (TIP3P, Qiao et al., *Nat. Commun.* 2022, 13, 2461) to describe the water confinement. Simultaneously, the run time has been extended. We observed that alterations to the water model and extensions to the simulation period did not alter our initial findings. Also, these simulation adjustments yielded valuable additional information, which has been supplemented in the revised manuscript.

To enhance sampling for providing more comprehensive explanation, additional starting

configurations were constructed (Figure R5). Each system was simulated for 50 ns (doubled run time) to investigate the equilibrium state of the layered structures. The structural profile at the inception of the simulation (0 ns) reveals the discernible separation of the GO sheets, along with the initial assembly of B₁₂. This early configuration provides crucial insights into the initial state of our system. During the simulations, GO and B₁₂ changed their position and orientation to reduce steric hindrance. It was observed that the inserted B₁₂ in the interlayers positively improved the structural stability, which can be attributed to the interactions between B₁₂ and GO (Figure R5c). Compared with GO-B₁₂ (Figure R5b), GO-B₁₂-GO exhibited a distribution of B₁₂ with less over-stacking. Additionally, density profiles (Figure R6) show that the density of water at the solution-electrode interface for GO-B₁₂-GO is lower that compared to GO-B₁₂. The intercalated B₁₂ exhibits a less solvated structure than the surface-adsorbed B₁₂. Consequently, these findings emphasize the vital contribution of B₁₂ in bolstering structural stability. It also highlights the extent to which solvation dynamics are influenced by the specific layered configurations, elucidating the intricate interdependence between molecular architecture and solvation behavior within these systems.

Figure R5. Snapshots of MD simulations for the layered models of (a-c) GO, GO-B₁₂ and GO-B₁₂-GO with water molecules hidden for visualization.

Figure R6. Density profiles for (a) GO-B₁₂ and (b) GO-B₁₂-GO in water solution. GO-B₁₂: upper GO sheet supporting B₁₂ molecules (denoted as GO-1), lower GO sheet (denoted as GO-2). GO-B₁₂-GO: upper and lower GO sheet (denoted as GO-1 and GO-2).

To confirm the evolution of the layered structures, more structural properties were analyzed and supplemented. Root mean square deviation (RMSD) profiles were used to assess the structural stability and convergence of the system over time. By monitoring the GO sheets of GO-B₁₂ and GO-B₁₂-GO electrodes (Figures R7a and b), the RMSD exhibits a value of ~ 0.1 nm, indicating structures converged in 50 ns. To quantify the strength of the interactions between GO sheets, the non-bonded interaction energies were decomposed to the short-range Coulombic (Coul-SR) energy and short-range L-J (LJ-SR) energy. Even though the decomposed energy quantity is not a binding energy, it serves as a useful parameter for monitoring the system throughout the simulation process. The average short-range interaction energies remained relatively constant with minor variations (Figures R7c and d). In addition, the average interlayer spacing of ~ 1.3 nm in hydrated GO-B₁₂-GO electrode and ~ 0.4 nm in GO-B₁₂ electrode was observed (Figure R8), which is consistent with the density profile of GO sheets. Drawing from the aforementioned discussions, the simulation systems were rigorously validated by utilizing enhanced sampling methodologies, extending run durations, and implementing multi-property monitoring strategies. These approaches ensured the accuracy and reliability of our simulation results.

Figure R7. RMSD curves and the non-bonded interaction energy profiles of GO sheets for (a, c) GO-B_{12} and (b, d) $\text{GO-B}_{12}\text{-GO}$ electrodes during the simulation.

Figure R8. RMSD curves of B_{12} and interlayer spacing profiles of (a) GO-B_{12} and (b) $\text{GO-B}_{12}\text{-GO}$.

To gain insights into the diffusion and movement of DCA, mean square displacement (MSD) analysis was employed. In comparison to DCA in the GO-B_{12} assembly, the DCA in the $\text{GO-B}_{12}\text{-GO}$ assembly exhibits a smaller average diffusion coefficient (Figures R9a and b). This difference can be attributed to the faster adsorption of DCA onto the $\text{GO-B}_{12}\text{-GO}$ electrode. Moreover, the radial distribution function analysis revealed a lower coordination number of water molecules around DCA in the $\text{GO-B}_{12}\text{-GO}$ electrode compared to the GO-B_{12} electrode (Figures R9c and d). This indicates a less solvated structure of DCA in the $\text{GO-B}_{12}\text{-GO}$

electrode, which is consistent with the results of the MD simulations using the default SPC/E water model. Within the GROMOS force field, polar and aromatic hydrogens are explicitly represented. Meanwhile, aliphatic hydrogen and carbon atoms are amalgamated to form united atoms. This methodology allows for more efficient and expeditious simulations, without compromising the overall integrity of the system representation. On one hand, this approximation is considered reasonable for some simulation scenarios, given that aliphatic hydrogens are generally considered to make negligible contributions to the overall dynamics (Kanduč et al., *Nat. Commun.* 2017, 8, 14899; Guan et al., *Nat. Commun.* 2023, 14, 1016). In the GO-B₁₂-GO system, various types of hydrogen atoms can be found in majority of the polar components, such as H₂O molecules, –OH, –COOH and –CONH₂ groups. This may help to explain the consistent results obtained from MD simulations using two different water models. On the other hand, it is worth mentioning that this simplification may limit the accuracy in describing interactions that specifically involve aliphatic hydrogen. In these instances, water models that include explicit aliphatic hydrogen are particularly significant, as they enable a more precise analysis of the specific contribution from aliphatic hydrogen (Rohaim et al., *Nat. Commun.* 2022, 13, 1574; Li et al., *Nat. Commun.* 2023, 14, 1030).

Figure R9. (a, b) MSD curves of DCA for analyzing diffusion behavior in GO-B₁₂ and GO-B₁₂-GO, (c) radial distribution function of water oxygen around DCA and (d) coordination number of water around DCA at different positions for both GO-B₁₂ and GO-B₁₂-GO electrodes.

Response to Reviewer 2's Comments

This manuscript developed an enzyme-like GO-B₁₂-GO heterostructure electrocatalysts for selective electroreduction of sodium trichloroacetate (TCA). The catalytic center (vitamin B₁₂) of dehalogenase in organohalide-respiring bacteria was chosen as the molecular catalyst to construct the GO-B₁₂-GO heterostructure. The results showed that the electrocatalyst exhibits excellent dechlorination performance, which enhances reduction of intermediate dichloroacetic acid by 7.8 folds against that without sandwich configuration and can selectively generate monochloro-groups from trichloro-groups. Molecular simulations as well as DFT are used to reveal the mechanism. Overall, the design strategy in this work is novel, the experiments are performed in high quality, and the manuscript is written well. However, there are a few concerns as given below.

We really appreciate the referee's highly positive evaluation of our work, and are grateful to the referee for his/her comments and suggestions to help us further improve the quality of our manuscript. We have made all the revisions as suggested by the referee.

1. As the authors mentioned in the manuscript, all the purified reductive dehalogenases contain cobalamin cofactor (i.e., vitamin B₁₂ and its derivatives), it that the synthesized GO-B₁₂-GO heterostructure a kind of combination of GO and enzyme instead of mimic enzyme?

We thank the referee for his/her valuable question. Our work seeks to mimic the enzyme-like functions present in dehalogenases by creating GO-B₁₂-GO heterostructures. In our manuscript, when referring to the GO-B₁₂-GO heterostructure, we discuss the importance of the cobalamin cofactor (Vitamin B₁₂ and its derivatives), which is an integral component of all hitherto purified reductive dehalogenases. The synthesized GO-B₁₂-GO incorporates GO and B₁₂ cofactor, but does not encapsulate the full enzyme. In this sandwiched heterostructure, our hypothesis is that the two-dimensional GO can provide a specific spatial environment that enhances the catalytic capabilities of B₁₂. Two-dimensional materials with tunable compositions and interlayers may provide a platform that offers the opportunities of rationally arranging active sites and controlling catalytic microenvironment to ensure reaction rate and selectivity (Novoselov et al., *Science* 2016, 353, aac9439). The specific spatial environment, which is akin to a protein binding pocket formed by the peptide chain, serves as the basis for our GO-B₁₂-GO design. In essence, the construction of GO-B₁₂-GO draws inspiration from natural enzymes, but it has been simplified and tailored specifically for electrocatalytic water dechlorination.

2. The interface behavior of the synthesized GO-B₁₂-GO heterostructure is unclear which is needed to be more investigated.

We thank the referee for his/her insightful suggestion. In accordance with the referee's constructive suggestion, we have augmented our revised manuscript with additional data and analysis regarding the interface behavior. Specifically, we have examined and discussed the alterations in structural properties and the dynamics of interactions upon electrochemical treatment at the solution-electrode interface.

To reveal the structural differences in the interfacial processes, *in situ* ATR-FTIR spectroscopy measurements were conducted for GO, GO-B₁₂ and GO-B₁₂-GO electrodes. The

intercalated B₁₂ in the GO-B₁₂-GO electrode was found to facilitate the mediation of protons and electrons across layers, leading to the enhanced reduction compared to GO-B₁₂ (Figure R11 and 12).

To reflect the interfacial behavior during electrochemical experiments, *ex situ* characterizations of GO-B₁₂-GO electrodes were carried out. Small-angle X-ray scattering (SAXS) patterns were obtained to reveal the internal stacking arrangements of the GO-B₁₂-GO electrodes. Before electrochemical treatment, the scattering results indicate a random or isotropic distribution of GO components (Figure R10a). In contrast, the well-defined isotropic scattering at $\alpha = 90^\circ$ suggests the enhanced alignment of GO sheets and the formation of a layer-by-layer structure in GO-B₁₂-GO after electrochemical treatment (Figures R10b and c). These observations are in line with the findings reported in the previous study (Hegde et al., *Nat. Commun.* 2020, 11, 830).

Figure R10. SAXS patterns of GO-B₁₂-GO obtained before (a) and after (b) electrochemical treatments with (c) the azimuthal angle (α) plots.

3. Did the authors analyze the effects of different loading amounts of B₁₂ on the electrochemical catalytic performance?

We thank the referee for his/her thoughtful suggestion. Indeed, the loading amount of B₁₂ is crucial and displays an effect on its spatial distribution, manifesting in two predominant configurations—intercalated and surface-adsorbed ones. The intercalated B₁₂, located between GO sheets, provides a protective environment akin to enzyme active pockets, which may stabilize B₁₂ and promote its catalytic activity. The surface-adsorbed B₁₂ is more exposed to the electrolyte and could also contribute to the overall electrocatalytic performance.

In both the original and revised manuscript, we endeavored to fabricate electrodes with varying loading amounts of B₁₂. However, the inherent architecture of GO interlayer and the steric limitations of the B₁₂ molecule itself restrict the B₁₂ amount integrated within the GO framework. We have evaluated the loading amount of B₁₂ by UV-vis absorption spectroscopy. Determined by the absorption peak at 360 nm (Figure 2d and Supplementary Figure 13), a loading of 0.0576 mg cm⁻² was achieved for GO-B₁₂-GO while a lower content was obtained for GO-B₁₂ (0.0365 mg cm⁻²) or B₁₂ alone (0.0363 mg cm⁻²), suggesting that the sufficient loading of B₁₂ for GO-B₁₂-GO was ascribed to its intercalation into the inner space (Figure 2e). Increasing the amount of B₁₂ on the surface of electrode by adding B₁₂ to the electrolyte can form a second layer of B₁₂. However, the reduction current density did not increase (Supplementary Figure 29), suggesting that the more surface-adsorbed B₁₂ cannot contribute to the current density. Meanwhile, increasing the amount of intercalated B₁₂ is expected to enhance

the electrochemical performance. It is worth noting that the amount of GO should be increased simultaneously to accommodate the incorporation of B₁₂ molecules. Thus, optimizing this balance is crucial for maximizing the catalytic efficiency of the GO-B₁₂-GO heterostructure.

4. The reason why the intercalated B₁₂ in the GO-B₁₂-GO electrode possesses a dramatically higher activity than the surface-adsorbed B₁₂ is unclear just based DFT or MD simulations. Some *in situ* characterizations such as *in situ* FTIR or *in situ* Raman are recommended to further analysis the mechanism of the reaction.

We thank the referee for his/her insightful suggestion. Accepting the referee's suggestion, we conducted in-situ ATR-FTIR spectroscopy to elucidate the effects of electrode configurations on the electrochemical reactions and understand the contribution of the intercalated B₁₂ to the enhanced activity. All electrodes were pretreated using electrochemical activation approach and dried in air before *in situ* experiments. In the case of GO alone electrode (Figure R11a), a significant increase of absorption peak at 1643 cm⁻¹ was detected upon absorption of water. This peak is assigned to the C=C skeletal vibrations of graphitic domains or the deformation vibration of intercalated water (Kumar et al., *Nat. Chem.* 2014, 6, 151–158). A decrease in peak area was observed under the electrochemical conditions with applied potential from -0.2 to -1.2 V vs. Ag/AgCl. Specifically, the absorption peaks at 1040 cm⁻¹ and 1392 cm⁻¹ exhibited a significant decrease at -0.8 V, suggesting the reduction of C-OH groups (Pei et al., *Nat. Commun.* 2018, 145). Simultaneously, a new peak at 1207 cm⁻¹ appeared at -1.0 V, indicating the formation of ethers. Furthermore, another new peak appeared with increasing intensity at 1566 cm⁻¹ from -0.8 V to -1.2 V, which is attributed to the formation of vibrational stretching of sp²-hybridized C=C, suggesting the formation of graphitic carbon components. It is worth noting that the peak intensity at 1643 cm⁻¹ also decreased, which can be ascribed to a loss of intercalated water (Acik et al., *ACS Nano.* 2010, 4, 5861–5868; Kumar et al., *Nat. Chem.* 2014, 6, 151–158). In the case of GO-B₁₂ and GO-B₁₂-GO, the characteristic peaks of B₁₂ were overlapped by the strong GO bands. Therefore, reduction of GO was employed as a probe reaction to reflect the differences between GO-B₁₂ and GO-B₁₂-GO electrodes under electrochemical conditions. As depicted in Figure R11b, the GO sheets undergo a much slower reduction in GO-B₁₂ compared to GO alone, suggesting that the electrode potential was applied to the surface-adsorbed B₁₂ molecules and reduction current was diverted to the reduction processes of Co^{III} → Co^{II} → Co^I. The GO-B₁₂ electrode exhibited a relatively unchanged peak at 1632 cm⁻¹, suggesting that the vibrations of C=C domains or the deformation vibration of intercalated water are remained. In contrast, the GO-B₁₂-GO showed a fastest reduction of GO sheets (Figure R11c), indicating that the intercalated B₁₂ in the GO-B₁₂-GO electrode facilitates the mediation of protons and electrons, leading to the enhanced reduction of GO sheets within the electrode. Therefore, according to the comparison of GO reduction in different electrode configurations, the high reduction activity of B₁₂ intercalated inside GO-B₁₂-GO was confirmed by the efficient transfer of protons and electrons across the layers.

Figure R11. *In situ* ATR-FTIR spectra of different electrode configurations (a) GO, (b) GO-B₁₂ and (c) GO-B₁₂-GO.

In addition to the reduction processes, the FTIR absorption peaks of water molecules show a high correlation with the microenvironments within various electrode configurations. From air phase to aqueous phase, the asymmetric wide peaks appeared between 2750 and 3750 cm⁻¹ can be characterized by five specific hydrogen-bonding (HB) configurations, including H₂O at ~3636 cm⁻¹, which lacks HBs; H₂O at ~3570 cm⁻¹ (DDA), which donates two HBs and accepts one; H₂O at ~3430 cm⁻¹ (DA), which donates one and accepts one HB; H₂O at ~3220 cm⁻¹ (DDAA), which both donates and accepts two HBs; and H₂O at ~3040 cm⁻¹ (DAA), which donates one and accepts two HBs (Liu et al., *Nat. Commun.*, 2021, 12, 6141; Bregante et al., *Nat. Catal.*, 2021, 4, 797–808). In the case of GO, based on the above analysis of GO reduction and restacking process, Figure R12a shows a loss of H₂O and a formation of -CH₂ groups (1850 and 1785 cm⁻¹). This can be attributed to the increased ratios of DA/DDA or DA/DDAA. Similar change of this peak was found in GO-B₁₂-GO with a more significant trend (Figure R12c). In the case of GO-B₁₂, the edge of this peak at the high wavenumber remained unchanged, suggesting that no loss of water in GO-B₁₂ and the -CH₂ peaks were relatively weak (Figure R12b). Therefore, the intercalated B₁₂ in GO-B₁₂-GO plays a critical role in facilitating the electrochemical reduction.

Figure R12. *In situ* ATR-FTIR spectra of H₂O in different electrode configurations of (a) GO, (b) GO-B₁₂ and (c) GO-B₁₂-GO.

5. In general, the catalyst in this work is neither a homogeneous nor a heterogeneous catalyst, but a coupling of both. It cannot provide the possibility of catalyst recovery with unlost catalytic activity, nor guarantee the long-term effectiveness of biological catalysts due to the environment-sensitive enzyme (CbFDH) is directly exposed to the reaction solution.

We thank the referee for his/her valuable comment. In the field of enzyme catalysis, optimization strategies have been developed in an effort to solve many practical problems, including limitations in enzyme stability, reaction efficiency and substrate selectivity (Chen et al., *Nat. Catal.* 2020, 3, 203–213). To address the challenges in achieving high stability of the environment-sensitive enzyme, a host–guest supramolecular strategy was developed to combine abiotic and biotic catalysts, exhibiting high activity and stability during an 18-hour reaction process (Zhao et al., *Nat. Commun.* 2020, 11, 2903). Similarly, an enzyme-encapsulating strategy was developed to entrap enzymes for applications in environments unfavorable for native enzymes, which allowed access of substrates to encapsulated enzymes while maintaining the protection to the enzymes (Hu et al., *Sci. Adv.* 2020, 6, eaax5785). In this work, the layered structure of GO-B₁₂-GO electrode provides a co-loading of the intercalated B₁₂ as well as the surface-adsorbed B₁₂ with tunable local environment. The long-term effectiveness of GO-B₁₂-GO electrode was supplemented with the structural stability (Figure R21) and the electrochemical performance of a 15-hour electrolysis (Figure R22). The results show the coordination structure of Co-N_x site was well-defined and maintained during the electrolysis at –0.9 V vs. Ag/AgCl. Furthermore, the catalytic activity of B₁₂ in GO-B₁₂-GO electrode exhibited efficient reductive dechlorination during the 15-hour electrolysis. Therefore, GO-B₁₂-GO provides an inner environment for B₁₂ to enhance its activity and stability during electrochemical catalysis.

6. As the author emphasized that the sandwich-like configuration of GO-B₁₂-GO heterostructure played an essential role in enhancing the performance, however, there is no obvious characterization result to prove the sandwich-like configuration.

We thank the referee for his/her thoughtful suggestion. To investigate the structural and chemical compositions across the layers in the sandwich-like configuration of GO-B₁₂-GO, synchrotron-radiation photoelectron spectroscopy (SRPES) measurements were employed. The analysis of chemical in-depth variations in ultra-thin organic layered structures is very challenging. SRPES has a key advantage of high energy resolution and has proven to be particularly useful for analyzing complex systems, such as surfaces, interfaces, films and nanostructures (Tao et al., *Science* 2008, 322, 932-934; Wang et al., *Nat. Commun.* 2023, 14, 3808). This method enables XPS depth profiling by changing the kinetic energy of the emitted photoelectrons. The C 1s spectra indicate the presence of C=C graphitic components and oxygen-containing groups (Figure R13), such as C=O groups from the side chain (–CONH₂) of B₁₂ molecules, as well as C–O and O–C=O groups of GO sheets (Voiry et al., *Science*, 2016, 353, 1413-1416; Chang et al., *Nat. Catal.*, 2022, 5, 222–230). At a constant level of photon energy, the atomic fractional ratio of carbon in different chemical states is determined by the ratio of peak areas. As shown in Figure R14, the ratio of C=O/C–O was 1.4 in GO-B₁₂ and 1.0 in GO-B₁₂-GO at a photon energy of 460 eV (at a depth of ~2.14 nm), suggesting the lower loading ratio of B₁₂ at the outer surface in GO-B₁₂-GO compared to that of GO-B₁₂. Moreover,

the ratio of oxygen-containing C and sp^2 C indicates a larger loading ratio of GO in GO-B₁₂-GO compared to GO-B₁₂. With an increase in photon energy up to 940 eV (at a depth of ~4.15 nm), the C 1s spectra show a shift of binding energy, suggesting the dual components of C=O/C–O at surface to the single component of C–O at subsurface. Thus, at the photon energy of 940 eV, a dominant component of GO was found in the layers below surface. Similarly, N 1s spectra also confirm the presence of –CONH₂ side chain from B₁₂ (Figure R15). Therefore, the spatial distribution of chemical components across layers range from 2~4 nm was confirmed by SRPES spectra in both GO-B₁₂ and GO-B₁₂-GO electrodes.

Figure R13. SRPES spectra of different electrode configurations recorded under the photon X-ray energies of 460, 580, 700, 820 and 940 eV, respectively. C 1s spectra obtained by SRPES for (a) GO, (b) GO-B₁₂ and (c) GO-B₁₂-GO.

Figure R14. C 1s SRPES spectra and the area ratio of different C states for (a) GO-B₁₂ and (b) GO-B₁₂-GO. (c) The area ratio of O states to C states.

Figure R15. N 1s SRPES spectra of (a) GO-B₁₂ and (b) GO-B₁₂-GO.

Response to Reviewer 3's Comments

In this study, the authors proposed and developed the GO-B₁₂-GO electrode for toward water dechlorination by mimicking the binding pocket configuration and catalytic center of reductive dehalogenases. The assembled heterostructure electrode by sandwiching a molecular catalyst into the interlayers of two-dimensional graphene oxide exhibits excellent dechlorination performance, which enhances reduction of intermediate dichloroacetic acid by 7.8 folds against that without sandwich configuration. Furthermore, the authors made efforts to clarify why and how the GO-B₁₂-GO works well for water dechlorination by means of DFT calculations and molecular simulations. The calculation results suggest that the sandwiched inner space plays an essential role in tuning solvation shell, altering protonation state and facilitating carbon–chlorine bond cleavage. Seemingly, the electrocatalytic performance of the catalyst the authors developed was good and the mechanism was proposed was interesting to researchers in the fields of science and technologies. However, taking a close look at the details of the manuscript, I wanted to more clearly understand the followings, which are unfortunately not described in the manuscript. I also have the following points that need to be better proven in the manuscript:

We really appreciate the referee's highly positive evaluation of our work, and are grateful to the referee for his/her comments and suggestions to help us further improve the quality of our manuscript. We have made all the revisions as suggested by the referee.

1: Whether the electronic structure of Co center in B₁₂ in the sandwich structure is the same as that of B₁₂ alone. This comparison is important for understanding the catalytic mechanism.

We thank the referee for his/her insightful suggestion. To investigate the electronic structures, DFT calculations were employed to conduct charge analysis. We find that the atomic charge for Co center in B₁₂ model varies with the electrode structures. Cobalt atomic charge of B₁₂ alone was 0.259 (Figure R16a). In addition, we employed GO layer with different composites to probe the effect of electrode configurations on the electronic structure of Co center in B₁₂ in the sandwich structure. Comparisons between the surface-adsorbed B₁₂ at nitrogen-doped grapheme (GN-B₁₂) and the intercalated B₁₂ in GN-B₁₂-GN exhibited an increase of atomic charge from 0.295 to 0.305 (Figure R16b). A similar increase of charge was determined for Co center in structures from GO-B₁₂ to GO-B₁₂-GO (Figure R16c). Moreover, the solvation structure at the electrolyte-electrode interface plays a critical role in running the electrochemical reactions (Hou et al., *Science* 2021, 374,172-178; Du et al., *Proc. Natl. Acad. Sci. USA* 2022, 119, e2214545119). Thus, water molecules were introduced to probe the effect of solvation structure on the electronic structure of Co center. According to the results of MD simulation, a simplified model of GO-B₁₂ with one layer of water was constructed for simulating the surfaced-adsorbed B₁₂ at the electrolyte-electrode interface. Moreover, a model of GO-B₁₂-GO with one water was built for the interacted B₁₂ with partial solvation structure. Figure R16d illustrates the atomic charge of Co atom, which was lower in GO-B₁₂-GO (0.326) compared to GO-B₁₂ (0.362). This result demonstrates the relationship between the sandwich-like electrode configuration and the solvation structures of the intercalated B₁₂, which significantly affects the electronic structure of Co center in B₁₂.

Figure R16. The optimized structures of the surface-adsorbed B_{12} and the intercalated B_{12} in the sandwich structure, (a) B_{12} alone, (b) nitrogen-doped graphene as support for GN- B_{12} and GN- B_{12} -GN, (c) oxygen-doped graphene as support for GO- B_{12} and GO- B_{12} -GO, (d) different solvation structures for GO- B_{12} and GO- B_{12} -GO. Atomic charge of Co atom was denoted as q_{Co} .

2: In addition to GIXRD, more experimental data are needed to support the GO- B_{12} -GO structure.

We thank the referee for his/her thoughtful suggestion. In addition to GIXRD, various techniques including Raman spectroscopy (Figure R1), FTIR spectroscopy (Figure R3), small-angle X-ray scattering (SAXS, Figure R10), and synchrotron-radiation photoelectron spectroscopy (SRPES, Figures R13-15) were employed to obtain additional experimental data. These complementary results provide valuable information for characterizing the structure of GO- B_{12} -GO and further understanding its properties. To provide visual images of the electrode configurations at the micrometer level, scanning electron microscopy (SEM) cross-section measurements were performed. SEM analyses confirmed the layered structures of GO, GO- B_{12} and GO- B_{12} -GO electrodes (Figure R17). Considering the detection level of energy-dispersive X-ray spectroscopy (EDS), the carbon and oxygen elements were utilized to represent the GO sheets, while the additional nitrogen element was used to indicate the presence of B_{12} molecules. Nitrogen was predominantly observed in the upper surface of GO- B_{12} , showing a significant decrease in nitrogen content toward the cross-section. In contrast, GO- B_{12} -GO exhibited a more uniform nitrogen distribution throughout the entire thickness, including the surface and the interior regions. Therefore, the layered structures of GO- B_{12} and GO- B_{12} -GO were further confirmed by elemental mapping.

Figure R17. SEM cross-sectional images of GO, GO-B₁₂ and GO-B₁₂-GO electrodes. Elemental mapping images by EDS showing the distributions of C, O and N, respectively.

3: Why do you think that the difference in performance of B₁₂ and CoPc comes from the difference in the interaction of DCA with B₁₂ and CoPc, but is related to the difference in spatial structure between GO-CoPc-GO and GO-B₁₂-GO?

We thank the referee for his/her valuable question. In the electrolysis experiments, GO-CoPc-GO exhibited a lower reductive dechlorination ratio compared to GO-B₁₂-GO. The origin of the difference in performance could be analyzed in two aspects. Firstly, we conducted a comparison of the intrinsic activities of B₁₂ and CoPc by calculating the interactions between DCA and the Co sites of the two molecular catalysts. Secondly, as suggested by the referee, a comparison of the spatial structures should be undertaken for the two layered structures. A dominant peak at 8.127° indicates that the interlayer distance of GO-B₁₂-GO was 1.088 nm (Figure 2c), while various interlayer distances ranging from 1.23 to 0.47 nm was determined for GO-CoPc-GO (Figure R18). Based on the characterizations of the interlayer structure of GO-B₁₂-GO and GO-CoPc-GO, the structural difference can be attributed to the difference in performance.

4: What is the interlayer distance of the GO-CoPc-GO electrode?

We thank the referee for his/her insightful question. XRD patterns indicate that the GO-CoPc-GO sheets had peaks at 7.16°, 9.36°, 18.38° and 18.78°, corresponding to an interlayer spacing ranging from 1.23 to 0.47 nm (Figure R18). The variations in interlayer distances were attributed to the π-π stacking arrangement of CoPc molecules. This conclusion is supported by snapshots obtained from molecular dynamics (MD) simulations (Supplementary Figures 24a and c).

Figure R18. XRD pattern of GO-CoPc-GO electrode.

5: In Supplementary Fig. 13, Two CV curves correspond to without TCA and with TCA, in the figure, what does means of “arrow and adding B₁₂ in the bulk”?

We thank the referee for bringing this to our attention. A detailed scheme of the cyclic voltammetry (CV) experiments was presented in Figure R19. As indicated by the arrow, an increased current density was observed in the TCA electrolyte (deep red line) compared with the blank electrolyte (deep blue line), indicating the catalytic activity of the GO-B₁₂-GO electrode in reducing TCA. Upon adding B₁₂ to the electrolyte (light blue line), B₁₂ could diffuse from the bulk electrolyte to the electrode surface driven by a concentration gradient, resulting in the formation of a second layer of B₁₂. However, it was observed that the reduction current density of TCA did not increase with the increasing number of surface-adsorbed B₁₂ (light red line). Therefore, the more surface-adsorbed B₁₂ cannot contribute to the current density, and the intercalated B₁₂ plays a key role in driving reduction reaction.

Figure R19. CV profiles of GO-B₁₂-GO electrode at a scan rate of 200 mV s⁻¹. The components of the electrolyte varied from blank solution to the addition of TCA or B₁₂ as well as the co-addition of TCA and B₁₂.

6: In addition to the role of solvent, is the adsorption of TCA molecules and reaction intermediates on B₁₂ also affected by space? How does the change in adsorption affect the reaction process?

We thank the referee for his/her insightful questions. To investigate the effect of adsorption process on the reaction, DFT calculations were conducted on GO-B₁₂ and GO-B₁₂-GO structures. The component of water was excluded from both systems, i.e., without involvement of solvation structures. DCA molecule, as the key intermediate of TCA dechlorination reaction, was employed to probe the effect of space on the adsorption process. The adsorption of DCA at GO-B₁₂ showed an energy change of -0.457 eV, suggesting a favored binding of DCA (Figure R20a). Additionally, a stronger binding of DCA to the interlayer space of GO-B₁₂-GO (-0.903 eV) was observed, potentially contributing to its higher catalytic performance (Figure R20b). Upon adsorption of DCA, a structural change of GO sheet was observed in GO-B₁₂-GO. This result indicates that the adsorption process can be enhanced by the interlayer space, which further facilitates the dechlorination reactions.

Figure R20. The optimized structures of the key intermediate (DCA) adsorbed on the B₁₂ site in (a) GO-B₁₂ and (b) GO-B₁₂-GO structures. Adsorption energy was denoted as E_{ads} .

7: It would be better for the authors to show the stability of the structure and performance of catalyst or electrode.

We thank the referee for his/her insightful suggestion. The stability measurement of GO-B₁₂-GO during the electrochemical dechlorination process has been supplemented in the revised manuscript. The results show that the electrochemical dechlorination performance and the Faradaic efficiency of dechlorination could be maintained after 15-hour electrolysis (Figure R21, i.e., Supplementary Figure 19). The local coordination environment of Co centers was determined by the Co K-edge extended X-ray absorption fine structure (EXAFS) and X-ray absorption near-edge structure (XANES) spectroscopies. XANES spectra suggest that the oxidation states for GO-B₁₂-GO before electrolysis was larger compared to that after electrolysis (Figure R22a, i.e., Supplementary Figure 20).

In the case of GO-B₁₂-GO before electrolysis, the sharp pre-edge peak at 7710 eV can be attributed to the penta-coordinated local structures of atomically dispersed Co-N_x sites. Additionally, a rising-edge peak at approximately 7714 eV was observed, corresponding to the square-planar symmetry (D_{4h}) of the Co-N₄ structure (Chang et al., *Energy Environ. Sci.* 2012, 5, 5305-5314; Lien et al., *Nat. Commun.* 2020, 11, 4233).

For GO-B₁₂-GO after electrolysis, the persistence of the pre-edge peak at 7710 eV indicates that the penta-coordinated Co sites were retained with B₁₂. The EXAFS analysis shows that GO-B₁₂-GO revealed a dominant peak at the Co–N coordination, without the formation of Co–Co bonds (Figure R22b). Therefore, the original configuration of Co centers in GO-B₁₂-GO is well maintained after electrolysis.

Figure R21. (a) The concentration ratios of TCA, DCA, MCA and Cl⁻, monitored during the 15-hour electrolysis at the GO-B₁₂-GO electrode. Reaction conditions: an applied potential of -0.9 V vs. Ag/AgCl, pH 7.0, and carbon fiber paper as the working electrode. (b) The current-time curves and Faradaic efficiency of the GO-B₁₂-GO electrode, recorded during a 15-hour electrolysis.

Figure R22. (a) Co K-edge XANES experimental spectra of GO-B₁₂-GO electrode before and after electrolysis, CoO, Co₂O₃ and Co foil. (b) Fourier transform of the EXAFS spectrum (k^3 -weighted) in R space. (c, d) Fourier transform of EXAFS spectrum in k space.

REVIEWERS' COMMENTS

Reviewer #1 (Remarks to the Author):

The authors did an excellent revision of their paper including additional experiments and modelling. The quality and depth of the analysis is significantly improved. Recommend publication in Nat Comms.

Reviewer #2 (Remarks to the Author):

The authors answered all of the questions and modified the manuscript, I recommend it be published.

Reviewer #3 (Remarks to the Author):

Revisions were correctly made on the manuscript, the authors provided a comprehensive response to reviewers. I believe that this is commendable.